# The Role of Sleep Disturbance, Depression and Anxiety in Frail Patients with AF–Gender Differences

**DOI:** 10.3390/jcm10010011

**Published:** 2020-12-23

**Authors:** Beata Jankowska-Polańska, Jacek Polański, Krzysztof Dudek, Agnieszka Sławuta, Grzegorz Mazur, Jacek Gajek

**Affiliations:** 1Department of Clinical Nursing, Wroclaw Medical University, 50-367 Wrocław, Poland; 2Department of Internal Medicine, Occupational Diseases, Hypertension and Clinical Oncology, Wrocław Medical University, 50-367 Wroclaw, Poland; jacek.polanski@umed.wroc.pl (J.P.); agnieszka.slawuta@umed.wroc.pl (A.S.); grzegorz.mazur@umed.wroc.pl (G.M.); 3Faculty of Mechanical Engineering, Technical University of Wroclaw, 50-370 Wrocław, Poland; krzysztof.dudek@pwr.edu.pl; 4Department of Emergency Medical Service, Wroclaw Medical University, 50-367 Wrocław, Poland; jacek.gajek@umed.wroc.pl

**Keywords:** sleep disturbance, depression, anxiety, FS, atrial fibrillation, gender

## Abstract

The aim of the study was to assess the link between anxiety and depression and frailty syndrome (FS) in patients with atrial fibrillation (AF) with regard to gender differences. Material and methods. The study was conducted on 158 patients with AF (mean age 70.4 ± 7.6). The study used the hospital anxiety and depression scale (HADS-M), the Athens insomnia scale (AIS) and the Edmonton frailty scale to assess and compare anxiety, depression, and sleep disturbance between frail and non-frail patients with AF. Results. FS was diagnosed in 53.2% of patients. A comparative analysis showed a statistically significantly higher severity level of anxiety (12.0 ± 2.6 vs. 8.4 ± 2.5, *p* < 0.001) and depression (12.5 ± 2.5 vs. 7.2 ± 3.3, *p* < 0.001) in frail patients compared to non-frail patients. The analysis of the level of anxiety, depression and FS did not show any significant differences between the studied women and men. However, statistically, significant differences were observed when FS occurred, regardless of gender. Anxiety disorders were observed in 75.5% of patients with FS and in 16.7% without frailty, whereas depressive disorders were observed in 73.6% of frail patients and in 4.2% without frailty. In an analysis of the impact of cumulative variables on the level of frailty, the risk of FS in patients with anxiety/depression and sleep disturbance is almost 500 times higher compared to patients without anxiety/depression and sleep disturbance. The risk of frailty in patients with sleep disturbance only is thirteen times higher than in the reference group, i.e., in patients without depression/anxiety and sleep disturbances. Conclusions: Patients with AF and FS show deeper anxiety, depression and sleep disturbances. Gender does not influence the risk of frailty in AF patients. Frailty in patients with AF is associated with a higher risk of depression, sleep disturbances and anxiety.

## 1. Introduction

Approximately one in four hospitalized elderly patients have atrial fibrillation (AF) [1]. The condition is more common with age, affecting 17% of 80-year-olds. AF is an independent risk factor for stroke, heart failure, mortality, and lower quality of life [2]. In addition to somatic problems relating to the symptoms of heart failure, many patients complain of deteriorated mood and sleep disturbance. The feeling of palpitations relating to an irregular heart rate may cause significant problems falling asleep, especially in those patients who prefer to sleep on their left side. Moreover, the lying position during sleep may increase pulmonary congestion and dyspnea, which may not only cause a patient to wake up but may also make sleep more shallow. The co-occurrence of those problems with reduced exercise tolerance may result in a deterioration of mood, which may initially manifest itself as anxiety and may progress, especially in vulnerable patients, to depression [3]. Sleep-disordered breathing, which is more common in patients with AF and poor sleep quality, appears to be an important risk factor for atrial fibrillation, but also fatigue, mood disorders, which is associated with the severity of disease symptoms and the quality of life [4,5,6].

Frailty is an important prognostic factor in older adults with cardiovascular diseases. Frailty syndrome (FS) is known as depletion of reserves, is a consequence of a decrease in physiological reserves of many organs. Moreover, FS is associated with poorer functioning in the biopsychosocial sphere, which translates into worse responses to stressors, both physical and psychological. The factors involved in the pathogenesis of frailty include biological, clinical, and social factors. The consequence of FS is the poorer functioning of elderly people in many spheres, as well as increased risk of illness, hospitalization, institutionalization and even death [7,8]. Additionally, AF is described as an independent marker of frailty. The severity of the symptoms of AF permitted higher levels of FS to be predicted. There is a growing number of patients with frailty and AF. FS was diagnosed in 25% of patients with cardiac arrhythmias, and a further 40% had a higher risk of frailty [9]. Frailty results in the loss of homeostatic reserve and immunity. It also leads to cognitive decline and higher sensitivity to stressors. This may result in an increased prevalence of anxiety and depression. While there are several hypotheses indicating an association between sleep disturbance and frailty in elderly people, the conclusions reached by the few studies available vary. However, the consistent evidence as to the correlation between the perceived quality of sleep and frailty suggests a possible bidirectional relationship between them, with clinical emphasis placed on sleep–wake disturbances as modifiable causative factors that contribute to FS. Some studies indicate significant gender differences. There are also studies that indicate that depressive symptoms are associated only with lower overall sleep satisfaction and that there is a link between anxiety symptoms and decreased overall sleep satisfaction, difficulty falling asleep, waking up at night, and also morning weakness and drowsiness during the daytime. Some studies suggest that the biochemical changes caused by sleep–wake disturbances, including changes in hypothalamus-pituitary-adrenal axis and hypothalamic-pituitary-gonadal axis functions, lower cortisol reactivity, lower levels of growth hormone and insulin-like growth factor-1 and chronic inflammation, etc., also play a part in the pathophysiology of frailty in older adults [5,6,10]. Sleep disturbance is correlated with the overall perception of health and may indicate the presence of health problems (cardiovascular problems, disability and depression), which may themselves be considered as predictors of FS [11]. Complaints of disrupted sleep are common in elderly people, with a prevalence reaching over 60% [12]. In light of all this information, it should be assumed that the population of elderly AF patients is particularly at risk of sleep disturbance, anxiety, depression and FS. All the described factors, when present, increase their mutual intensification. There were few studies discussing the relationship between sleep disturbance and depression and anxiety in community-dwelling individuals. A few studies have investigated the link between sleep disturbance and depression in elderly patients [13]. However, there is little information available on the relationship between sleep disturbance and anxiety and depression among elderly frail patients [12].

In view of the above, it is necessary to assess the level of sleep disturbance and depressive symptoms in AF patients as well as the link between the disturbances and the prevalence of FS.

## 2. Aim

The aim of the study was to assess the prevalence of sleep disturbance as well as depressive and anxiety disorders in an unselected group of patients hospitalized for AF. Moreover, we aimed to establish the relationship between the aforementioned disorders and frailty in those patients. The study also examined differences between male and female patients.

## 3. Material and Methods

Inclusion criteria were as follows:

Confirmed diagnosis of AF as per the EHRA criteria, age ≥ 60 years, consent to participate, cognitive function sufficient to complete the questionnaire unassisted.

Exclusion criteria were as follows:

Age < 60 years, lack of consent or withdrawal of consent during the study, cognitive impairment indicating dementia or interfering with the understanding and completion of surveys, depression and anxiety requiring pharmacological treatment.

All 158 patients provided informed consent to participate in the study, and their clinical condition was stable. The patients were informed that participation in the study was strictly anonymous and that they could withdraw from it at any time without providing justification.

The study was approved by the Bioethics Committee of the Wroclaw Medical University.

The study used a diagnostic survey, including the following instruments:

### 3.1. Data Collection

The authors’ own questionnaire recording information on the patient’s gender (F, M), age, education, marital status and residence. Clinical data were obtained from clinic files.

### 3.2. Frailty Assessment

The Edmonton frailty scale (EFS) includes 10 domains relating to cognitive functions, mobility, balance, mood, social support, nutrition, health attitudes, quality of life (QoL), medication and functional independence. The geriatric condition evaluation is determined by physical, psychological and social aspects. Each item is scored between 0 and 3. The overall maximum score is 17 and represents the highest level of frailty [14].

### 3.3. Psychological Instrument for the Assessment of Anxiety and Depression

In order to determine the prevalence of anxiety and depression, we used the hospital anxiety and depression scale (HADS). It is different from all other scales in that it makes it possible to assess anxiety and depression without investigating somatic symptoms. HADS is often used to analyze different conditions in a clinical setting. The scale was originally developed by Zigmond and Snaith (1983) and includes 14 questions, seven of which relate to anxiety (HAD-A), while the other seven relate to depression (HAD-D). The authors of the scale considered a score of less than 8 to suggest that no mental disorder was present, a score equal to or greater than 8 to indicate that a mental disorder was probably present, and a score greater than 10 to indicate that the patient was highly likely to have a mental disorder [15].

### 3.4. Assessment of Sleep Disturbance

The Athens insomnia scale (AIS) is a short, eight-item scale enabling a quantitative assessment of the symptoms of insomnia based on the ICD-10 criteria. The original validation studies showed the high reliability and accuracy of the tool. The overall score ranges from 0 to 24, with a higher score indicating poorer quality of sleep. A score of 6 or higher was considered to indicate, with high probability, the presence of insomnia (sensitivity of the scale—93%, specificity of the scale—85%) [16]. The AIS is one of the most commonly used scales for the diagnosis of insomnia and for the assessment of the effectiveness of treatment for insomnia [17].

### 3.5. Statistical Analysis

For statistical analysis purposes, the data collected in the study were recorded, processed and analyzed using Statistica 13.3 (TIBCO Sotfware Inc., Palo Alto, CA, USA). The statistical analyses of the survey data included the following steps: qualitative variables measured on the nominal (e.g., gender) and ordinal scales (e.g., education) were cross-tabulated. The strength of the correlations between the pairs of variables was evaluated using the chi-squared test. Fisher’s exact test was used where the expected count in at least one cell of a four-field table was lower than 5. Means (M), standard deviations (SD), medians (Me), lower quartile (Q1) and upper quartile (Q3) values, as well as ranges (minimum and maximum), were calculated for all quantitative variables. For quantitative variables (e.g., age), the normality of distribution was verified using the Shapiro–Wilk test. The homogeneity of variance was assessed using Bartlett’s and Levene’s tests. A Student’s *t*-test was used to assess the significance of differences between the mean values of variables with normal distribution and homogeneous variances in two independent groups. The nonparametric Mann–Whitney test was used to verify the significance of differences between the mean values of variables with non-normal distribution or with heterogeneous variances in two groups. Regression analysis based on Pearson’s r linear correlation coefficient was used to determine the strength and direction of linear correlations between two continuous variables. The least-squares method was used to estimate regression coefficient values. Multiple regression was used to assess the impact of the factors (predictors) analyzed on the dependent variable. In multiple regression models, regression coefficients b_i_ were standardized to make their values independent of the value range of the random variable associated with the coefficient (standardized regression coefficients b_i_ range between −1 and +1, and may be compared for different random variables; the higher the absolute value of the standardized regression coefficient, the stronger the impact of the variable on the dependent variable). A level of significance of *p* = 0.05 was used for all statistical analyses. The results of the statistical analyses are shown in graphs or tables.

## 4. Results

### 4.1. Characteristics of the Study Population

The study included 158 patients (mean age 70 ± 7 years) treated for AF. For the purpose of analysis, the patients were divided into two groups by gender to identify possible differences between male and female respondents. Group 1—women, *n* = 78; Group 2—men, *n* = 80. The basic statistics for parameters characterizing study participants are included in Table 1. The only differences observed between male and female respondents were in the level of education and type of work performed.

Clinical characteristics: Differences in symptoms were observed between male and female respondents. More male respondents than female respondents suffered from palpitations (80.0% vs. 56.4, *p* = 0.003), whereas more female patients than male patients were diagnosed with heart failure (28.2% vs. 13.8%, *p* = 0.041) and hyperthyroidism (26.9% vs. 11.2%, p 0.021). Differences between male and female patients were also observed as regards the frequency of taking anti-AF drugs: female patients were instructed to take the drugs prescribed twice a day (55.1% vs. 32.5%), whereas male patients were instructed to take drugs once a day (67.6% vs. 44.9%). However, more male patients than female patients experienced bleeding symptoms as a result of the treatment used. The results are presented in Figure 1 and Table 2.

### 4.2. Assessment of the Anxiety and Depression

Men and women presented a borderline abnormal level of anxiety and depression according to the HADS questionnaire. In addition, a low level of disease acceptance, a high level of sleep disturbance was observed in the study group, and only 41% of women and 32% of men did not have FS. A comparative analysis between males and females did not show any differences in the level of anxiety and depression, sleep disturbance and the prevalence of FS. The results are presented in Table 3.

In the study group, 48.1% had anxiety disorders, whereas 39.9% had depressive disorders. Significantly more male patients than female patients had anxiety disorders (55.0% vs. 41.0%; *p* < 0.001).

### 4.3. Depression, Anxiety and Sleep Disturbance in AF Patients

In the next step, comparative analyses of the level of anxiety and depression were performed in groups depending on gender and the presence of FS. The results of the analysis showed statistically significant differences between the respondents. Both male and female patients with FS had a statistically significantly higher level of anxiety, depression and sleep disturbance. In the group of women with FS, as many as 65.1% had an abnormal level of anxiety, 64.4% had an abnormal level of depression and 95.4% reported significant sleep disturbance. Similar results were found for male respondents: 82.9% had an abnormal level of anxiety, 70.7% had an abnormal level of depression, and 95.1% reported sleep disturbance. (Table 4)

For comparison, a similar analysis was performed for the entire group, considering only the presence of the FS. The results showed that a higher proportion of patients with FS compared to patients without frailty had an abnormal level of anxiety (73.8% vs. 18.9%), abnormal level of depression (72.6% vs. 2.7%) and a high level of sleep disturbance (95.2 vs. 37.8) (*p* < 0.001).

### 4.4. Factors Associated with Frailty Syndrome

Univariate analysis showed that high levels of anxiety, depression and sleep disturbance were associated with FS.

In order to determine the optimal cutoff values for quantitative parameters (age, duration of AF, etc.), an analysis of receiver operating characteristics (ROC) curves was carried out using Youden’s statistics.

Chi-squared test results showed that in the case of female and male patients with AF, there was a significant correlation between frailty and such parameters as age > 65, lack of education, inactivity in the labor market, living alone, lack of physical activity and duration of AF of more than 5 years. Moreover, in the two groups studied, the risk of frailty was higher among patients with a history of fatigue, palpitation, dizziness, fainting, diabetes, heart failure, stroke/TIA, frequent hospital stays, bleeding as a result of treatment and the previously analyzed depression, anxiety and sleep disturbances. The results are shown in Table 5.

The estimated odds ratio (OR) indicates that the likelihood of developing FS is almost 100 times greater in depressed patients than in non-depressed patients (OR = 95.5), 33 times greater in patients with sleep disorders than in patients without sleep disorders (OR = 32.9) and 23.4 times higher in patients with anxiety. In the case of the remaining analyzed variables, a high probability of FS was observed in patients without higher education (OR = 59.8), physically inactive (OR = 48.2) and older (OR = 37.6).

### 4.5. Possible Predictors of Frailty Syndrome

An analysis of the impact of cumulative variables on the level of frailty was carried out, showing that the risk of FS in patients with anxiety/depression and sleep disturbance is almost 500 times higher (OR = 495) compared to patients without anxiety/depression and sleep disturbance. The risk of FS in patients with sleep disturbance only is thirteen times higher (OR = 13.3) than in the reference group, i.e., in patients without depression/anxiety and sleep disturbance.

Logistic regression coefficients for FS and the sociodemographic and clinical characteristics analyzed showed that independent determinants of FS in patients with AF are as follows: duration of AF > 5 years (β = 4.4649; *p* = 0.001), anxiety (β = 2.4727; *p* = 0.036), depression (β = 5.571; *p* < 0.001) and sleep disturbance (β = 4.1052; *p* + 0.014).

A model with significant parameters only:

Logit(p) = −8.54 + (4.649 × duration of AF > 5 years) + (2.473 × anxiety) + (5.571 × depression) + (4.105 × sleep disturbance).

Likelihood ratio test: overall model fit: chi-squared = 152.3; DF = 4; *p* < 0.0001.

The strongest determinant of FS is depression (OR = 262.7), followed by a duration of AF of more than 5 years (OR = 104.5). The third and fourth strongest predictors are sleep disturbance and anxiety. The results are presented in Table 6.

## 5. Discussion

The most featured results of our study indicate a close relationship between the presence of arrhythmia on one side and the sleep disorders, depression signs and FS. It is surprising that there is such a scarcity of studies on this issue. In the light of the literature, it seems that researchers focus on the strictly medical aspects of the condition, such as treatment, prevention of AF episodes, anticoagulation therapy and surgical treatment of arrhythmia, underestimating its emotional and psychological aspects [18]. Sleep disturbance, anxiety attacks and deterioration of mood significantly impair the physical and social functioning of AF patients, which is associated with, and increases, frailty [19].

The majority of patients participating in our study were at risk of frailty or had FS. It must be stressed that FS was more prevalent in male patients with AF, even if the difference was not statistically significant. There is an ongoing discussion in the literature on the association between AF and FS. However, there is still no uniform definition of FS and the factors that may contribute to frailty. A meta-analysis by Villani et al., including 11 studies published between 2002 and 2017, showed conflicting results, as only 4 studies confirmed a strong correlation between AF and FS [20]. Another meta-analysis also showed that there is a lack of evidence regarding the link between frailty, AF, anticoagulation and clinical outcomes [21]. However, a clinical practice strongly indicates that there is a correlation between AF and older age, which in turn is clearly associated with a higher prevalence of FS. The prevalence of AF in people aged over 80 is between 10 and 25%, which is similar to the prevalence of frailty [22].

Under physiological conditions, people are not aware of their heartbeat. Therefore, an irregular heart rate and the associated varying stroke volume result in a constant sensation of a quivering heartbeat in patients with AF. With a few rare exceptions, more or less severe heart palpitations are an inherent symptom of AF [23,24]. Palpitations are more acutely experienced by women, slim individuals and persons lying on the left side [25,26]. Some patients report that the sensation of palpitations becomes less intense and disappears with the duration of sustained arrhythmia. However, a large proportion of patients report palpitations as a permanent symptom. In addition, in patients with paroxysmal arrhythmia, the symptoms are even more profound, as these patients can compare their recent state of full health with their current condition.

One important aspect of the pathology concerned and its symptoms is the hemodynamic condition of patients during AF [27]. Relatively slow, pharmacologically controlled, atrioventricular conduction is associated with more arrhythmia tolerance and a lack of signs of pulmonary congestion, i.e., the symptoms of heart failure. This is often observed in patients with chronic arrhythmia. In turn, the episodes of AF are usually associated with fast, poorly controlled, atrioventricular conduction. Therefore, in patients with paroxysmal AF, the symptoms of heart failure, including in particular dyspnea and anxiety, occur more frequently and are more pronounced. This situation may occur in patients with chronic AF for various reasons, especially when they stop taking their medication. AF is often accompanied by frailty, as the disease, in the vast majority of cases, affects older people and is associated with multiple comorbidities. The symptoms listed above obviously disrupt patient functioning, both during physical activity and during rest periods. The symptoms may be particularly pronounced at night when the surrounding silence and lack of other external stimuli cause the symptoms of arrhythmia to be experienced more acutely. Thus, sleep disturbance, restlessness and anxiety are common in patients with AF and accompany both paroxysmal and chronic forms of this arrhythmia [28].

Our results may indicate furthermore that all the above-mentioned factors could facilitate the development of FS or increase its severity. The logistic regression coefficients for FS and the sociodemographic and clinical characteristics analyzed in this study showed that independent determinants of FS include the duration of AF of more than five years, as well as the presence of depression, anxiety and sleep disturbance. However, an analysis of the impact of cumulative variables on the level of frailty showed that the risk of FS in patients with anxiety/depression and sleep disturbance is almost 500 times higher compared to patients without anxiety/depression and sleep disturbance. The risk of FS in patients with sleep disturbance only is thirteen times higher than in the reference group, i.e., in patients without depression/anxiety and sleep disturbance. This demonstrates that the issue discussed in our study is very important and that particular attention should be paid to those patients.

In summary, to the best of our knowledge, this is the first study to evaluate the relationships between the symptoms of anxiety, depression and complaints relating to sleep disturbance in a population of AF patients with FS. Our study showed that the level of mood disorders in AF patients is very high and that the symptoms affected the majority of the patients studied. It must be stressed that male patients participating in our study had a significantly higher level of anxiety compared to female participants. This is in line with some literature reports. The unpredictable nature of this type of arrhythmia, characterized by an unexpected onset or recurrent events, has a significant impact on psychological disturbances as compared with the general population [29].

The levels of anxiety and depression increase with recurrent AF episodes and are associated with the severity of symptoms [30,31]. Anxiety is the most common response to a diagnosis of AF in patients without associated comorbidities [32]. The higher level of anxiety in male patients participating in our study may be explained by a significantly higher intensity of palpitation symptoms. While clinical experience shows that men tolerate sustained AF better than women, it is possible that in men, who are used to a physiologically slower heart rate, the presence of palpitations leads to a higher level of anxiety and sleep disturbances. However, if not asked directly, men are unlikely to raise this issue during a visit to a physician, especially when the physician focuses on such aspects as the maintenance of sinus rhythm and the proper control of ventricular rhythm. A comparative analysis of mood and sleep disorders between male and female patients with and without FS, as carried out in this study, showed that in both men and women with FS or at risk of frailty, the severity of those disorders was significantly higher compared to patients without FS. It is worth pointing out that scientific evidence shows that there is a significant association between sleep disturbance and the symptoms of depression and anxiety in elderly patients, regardless of the underlying condition [33,34]. Unfortunately, it is also known that treatment for AF may improve the severity of symptoms while not reducing the level of anxiety and depression. As pointed out above, this may stem from the failure to recognize the role of patients’ emotions or result from a different target outcome of the treatment used. This shows that our study is important in that it deals with an underestimated aspect of AF. Sleep disturbance, especially in patients with FS, results in a faster social and psychological deterioration of patients and increases the risk of death [31]. In patients with anxiety and sleep disturbance, there is a loss of the normal fall in blood pressure at night, which leads to a higher number of complications of hypertension [35]. In turn, chronic anticoagulation in AF patients increases the risk of bleeding complications.

A study by Press et al. of 496 elderly patients showed that there was a correlation between depression and sleep satisfaction and that anxiety was associated with decreased overall sleep satisfaction, difficulty falling asleep, waking up at night, and morning weakness as well as daytime drowsiness. According to the authors, sleep disturbances are very common in elderly frail people and are associated significantly more with anxiety than with depression [12]. The results of the aforementioned study confirm that the risk of complications relating to sleep disturbance is higher in AF patients, especially those with FS, which is clearly in line with our results.

Our study included a gender-sensitive analysis of factors associated with FS depending on the selected socio-clinical variables. A higher level of frailty was observed in patients who were of older age, uneducated and physically inactive, patients who lived alone, patients with more severe symptoms of AF and patients with comorbidities, especially those with diabetes and heart failure. The last can be in particular related to the tachycardiomyopathy mechanisms and could be cured only with relatively new implantation techniques [36,37]. Interestingly, our study did not show differences between male and female patients, and the variables analyzed had a statistically significant impact on the level of frailty in both male and female respondents. Most of those variables were also investigated in the available clinical studies on elderly patients. However, data regarding patients with AF is scarce. A study by Ding et al. confirmed the results of the previous study by Bergman et al. as regards the determinants of FS [38,39]. The potentially modifiable factors listed by the authors include low physical activity, cognitive disorders, depression and lack of social support. Moreover, low education, obesity and a low level of wealth may be addressable early or mid-life predictors. Furthermore, it was concluded that chronic diseases and social isolation might contribute to the development of FS. The multiplicity of factors associated with FS indicates an extensive mechanism for the development of this condition. The symptoms of AF and their negative impact on sleep disturbance, anxiety and depression, as shown in the present study, trigger a worrying vicious circle that results in a faster deterioration of the patient’s condition and increases the risk of death.

In light of the above, our results are of great clinical and practical significance. We point out an underestimated aspect of AF, which is often omitted in the standard clinical assessment of patients. Cardiologists, who focus on the maintenance of sinus rhythm and control of heart rate, monitor the difficult and dangerous anti-arrhythmic and anticoagulation treatment and assess the level of thromboembolic and bleeding risk, fail to ask patients simple questions regarding their emotions relating to arrhythmia and the quality of sleep. This failure may, in turn, result in complications that may shorten a patient’s life, as is the case for heart failure and thromboembolic stroke.

## 6. Conclusions

No difference was observed between male and female patients with AF in the severity of the symptoms of depression and sleep disturbance.

Patients with AF and FS show more severe symptoms of anxiety, depression and sleep disturbance compared with AF patients without frailty regardless of gender.

However, there is a significant correlation between the risk of frailty in male and female AF patients and older age, low level of education, inactivity in the labor market, lack of social and physical activity, and the symptoms of the disease and comorbidities.

Independent predictors of depression, sleep disturbances and anxiety in AF patients are FS and five years duration of AF, while FS increases the risk of depression, sleep disturbances and anxiety. The co-occurrence of depression, anxiety and sleep disturbance in AF patients is associated with an almost five hundred times higher probability of FS.

## 7. Practical Implications

The results of our study show how important it is to ensure that the clinical assessment of elderly AF patients includes an assessment of their psychological condition and sleep disturbance. Therefore, there is an urgent need to implement a screening program to identify the group of patients who are particularly at risk of mood disorders, sleep disturbances and FS and plan measures to reduce the risk or delay the progression of frailty through early diagnosis and treatment of depression and anxiety, as well as prevention of sleep disturbance.

## 8. Study Limitation

The present study has some limitations. According to the nature of the hospitalization of the AF patients, the arrhythmia was not classified into a paroxysmal, persistent or chronic form. Even if the symptoms of different forms of AF are quite similar, some of the patients can differently sense the paroxysms of AF and the increase of uncontrolled heart rate. The same applies to arrhythmia treatment. Our patients undergo pharmacological treatment aimed at achieving sinus rhythm maintenance and ventricular rhythm control as well. As the study group included patients of older age, there were no attempts to treat the patients invasively by means of pulmonary vein isolation. All those issues could influence the obtained results. Nevertheless, it does not make our conclusions and practical implications less important.

## Figures and Tables

**Figure 1 jcm-10-00011-f001:**
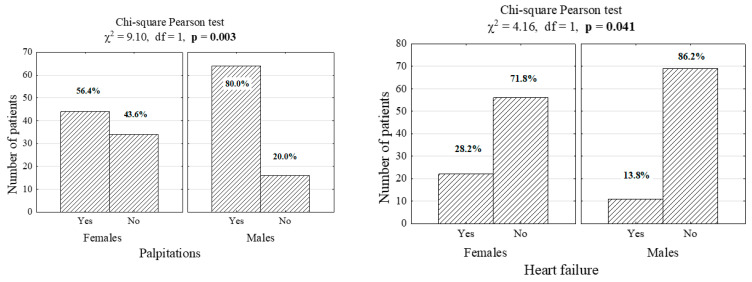
Number (percentage) of patients in gender and co-morbid groups (χ^2^—independence test statistics, df—degrees of freedom, *p*—significance level for the test).

**Table 1 jcm-10-00011-t001:** Basic statistics for sociodemographic parameters characterizing the population of 158 patients.

Variable	Females	Males	*p*-Value
	(*n* = 78)	(*n* = 80)	
	*n* (%)	*n* (%)	
Age, mean ± SD	71.8 ± 8.5	70 ± 7	0.346
Education			0.009
Primary	8 (10.3)	3 (3.8)	
Vocational	29 (37.2)	46 (57.5)	
Secondary	27 (34.6)	13 (16.3)	
Higher	14 (17.9)	18 (22.5)	
Source of livelihood			0.163
Occupationally active	22 (28.2)	32 (40.0)	
Not working (disability pension, retirement pension)	56 (71.8)	48 (60.0)	
Marital status			0.154
Single	36 (46.2)	46 (57.5)	
In a relationship	42 (53.8)	34 (42.5)	
Place of residence			0.435
Village	35 (44.9)	31 (38.8)	
City	43 (55.1)	49 (61.3)	
Type of job			0.017
Physical work	31 (39.7)	49 (61.3)	
Mental work	27 (34.6)	21 (26.3)	
Other work	20 (25.6)	10 (12.5)	
Physical activity			0.758
No	38 (48.7)	36 (45.0)	
Yes	40 (51.3)	44 (55.0)	

**Table 2 jcm-10-00011-t002:** Basic statistics for clinical parameters characterizing the population of 158 patients with atrial fibrillation (AF).

Variable	Females	Males	*p*-Value
	(*n* = 78)	(*n* = 80)	
Duration of AF (years), mean ± SD	7.6 ± 5.1	6.3 ± 4.1	0.078
Basic symptoms of the disease			
Dyspnea	55 (70.5)	51 (63.8)	0.462
Quick fatigue	50 (64.1)	45 (56.3)	0.398
Dizziness	41 (52.6)	31 (38.8)	0.113
Pain in the chest	25 (32.1)	26 (32.5)	0.912
No symptoms	19 (24.4)	16 (20.0)	0.640
Comorbidities			
Hypertension	61 (78.2)	65 (81.3)	0.781
Ischemic heart disease	41 (52.6)	40 (50.0)	0.870
Diabetes	43 (55.1)	48 (60.0)	0.647
Stroke/TIA	20 (25.6)	22 (27.5)	0.933
Number of hospitalizations			0.075
1–2 times	20 (25.6)	20 (25.0)	
3–5 times	17 (21.8)	31 (38.8)	
More than 6	41 (52.5)	29 (36.3)	0.141
More than 10 times	27 (34.6)	16 (20.0)	
Anticoagulant treatment			
NOAC	50 (64.1)	41 (51.3)	0.141
VKA	28 (35.9)	39 (48.8)	
Controls the INR indicator			
Yes	77 (98.7)	68 (85.0)	0.002
No	1 (1.3)	12 (15.0)	
Frequency of checking the INR indicator			
Every two months or more often	1 (1.3)	13 (16.3)	<0.001
Every three months or less often	32 (41.0)	27 (33.8)	
Frequency of taking anti-AF drugs			
Twice a day	43 (55.1)	26 (32.5)	0.003
Once a day	35 (44.9)	54 (67.6)	
Bleeding as a result of treatment	25 (32.1)	36 (45.0)	0.132

**Table 3 jcm-10-00011-t003:** Assessment of depression and anxiety symptoms and sleep disturbance in men and women using the hospital anxiety and depression scale (HADS) questionnaire and the Athens insomnia scale (AIS).

Hospital Anxiety and Depression Scale (HADS)	Females	Males	*p*-Value
	(*n* = 78)	(*n* = 80)	
HADS-anxiety (total score)			0.210
Mean ± SD	9.9 ± 2.5	10.6 ± 3.5	
HADS-depression (total score)			0.972
Mean ± SD	9.8 ± 4.1	9.8 ± 3.6	
Depression assessment			0.958
Normal (0–7)	17 (21.8)	18 (22.5)	
Borderline abnormal (8–10)	29 (37.2)	31 (38.8)	
Abnormal (11–21)	32 (41.0)	31 (38.8)	
AIS (total score)			0.210
Mean ± SD	15.2 ± 6.3	13.5 ± 5.8	
Me (IQR)	15 (9–23)	14 (8–16)	
Sleep disorder assessment			0.147
Normal (0–5)	0 (0.0)	3 (3.8)	
Borderline abnormal (6–10)	21 (26.9)	26 (32.5)	
Sleep disturbance (11–24)	57 (73.1)	51 (63.7)	
Edmonton frailty scale (total score)			0.787
Mean ± SD	8.7 ± 4.9	8.4 ± 5.1	
Me (IQR)	9 (4–14)	8 (4–13)	
Assessment of frailty syndrome level			0.100
Not Frail (0–5)	32 (41.0)	26 (32.5)	
Vulnerable (6–7)	3 (3.8)	13 (16.3)	
Mild frailty (8–9)	23 (29.5)	18 (22.5)	
Moderate frailty (10–11)	8 (10.3)	8 (10.0)	
Severe Frailty (12–17)	12 (15.4)	15 (18.8)	

**Table 4 jcm-10-00011-t004:** Depression, anxiety and sleep disturbance in atrial fibrillation (AF) patients depending on gender and the presence of frailty syndrome (FS).

Variable	Female		*p*-Value	Male		*p*-Value	All Patients		*p*-Value
	FS = Yes (*n* = 43)	FS = No (*n* = 35)		FS = Yes (*n* = 41)	FS = No (*n* = 39)		Yes (*n* = 84)	No (*n* = 74)	
	*n* (%)	*n* (%)		*n* (%)	*n* (%)		*n* (%)	*n* (%)	
Anxiety assessment (scores)			<0.001			<0.001			<0.001
Normal (0–7)	0 (0.0)	9 (25.7)		3 (7.3)	18 (46.2)		3 (3.6)	27 (36.5)	
Borderline abnormal (8–10)	15 (34.9)	22 (62.9)		4 (9.8)	11 (28.2)		19 (22.6)	33 (44.6)	
Abnormal (11–21)	28 (65.1)	4 (11.4)		34 (82.9)	10 (25.6)		62 (73.8)	14 (18.9)	
Depression assessment			<0.001			<0.001			<0.001
Normal (0–7)	0 (0.0)	17 (48.6)		0 (0.0)	18 (46.2)		0 (0.0)	35 (47.3)	
Borderline abnormal (8–10)	11 (25.6)	18 (51.4)		12 (29.3)	19 (48.7)		23 (27.4)	37 (50.0)	
Abnormal (11–21)	32 (74.4)	0 (0.0)		29 (70.7)	2 (5.1)		61 (72.6)	2 (2.7)	
Sleep disorder assessment			<0.001			<0.001			<0.001
Normal (0–5)	0 (0.0)	0 (0.0)		0 (0.0)	3 (7.7)		0 (0.0)	3 (4.1)	
Borderline abnormal (6–10)	2 (4.6)	19 (54.3)		2 (4.9)	24 (61.5)		4 (4.8)	43 (58.1)	
Sleep disturbance (11–24)	41 (95.4)	16 (45.7)		39 (95.1)	12 (30.8)		80 (95.2)	28 (37.8)	

**Table 5 jcm-10-00011-t005:** Factors associated with frailty syndrome (FS) in patients with atrial fibrillation (AF) and chi-squared test results.

Parameter (Risk Factor)	Female		*p*-Value	Male		*p*-Value
	FS = Yes (*n* = 43)	FS = No (*n* = 35)		FS = Yes (*n* = 41)	FS = No (*n* = 39)	
	*n* (%)	*n* (%)		*n* (%)	*n* (%)	
Age > 65 years	41 (95.4)	8 (22.9)	<0.001	37 (90.2)	11 (28.2)	<0.001
No higher education	43 (100.0)	21 (60.0)	<0.001	40 (97.6)	22 (56.4)	<0.001
Not working	43 (100.0)	13 (37.1)	<0.001	35 (85.4)	13 (33.3)	<0.001
Marital status—single	30 (69.8)	12 (34.3)	0.002	23 (56.1)	11 (28.2)	0.012
Physical work	27 (62.8)	4 (11.4)	<0.001	30 (73.2)	19 (48,7)	0.025
Lack of physical activity	35 (81.4)	3 (8.6)	<0.001	33 (80.5)	3 (7.7)	<0.001
Duration of AF > 5 years	40 (93.0)	9 (25.7)	<0.001	33 (80.5)	5 (12.8)	<0.001
Dyspnea	40 (93.0)	15 (42.9)	<0.001	27 (65.8)	24 (61.5)	0.688
Fatigue	35 (81.4)	15 (42.9)	<0.001	32 (78.1)	13 (33.3)	<0.001
Palpitations	18 (41.9)	26 (74.3)	0.004	26 (63.4)	38 (97.4)	<0.001
Dizziness	28 (65.1)	13 (37.1)	0.014	25 (61.0)	6 (15,4)	<0.001
Pain in the chest	19 (44.2)	6 (17.1)	0.011	17 (41.5)	9 (23.1)	0.079
Fainting	21 (48.8)	1 (2.9)	<0.001	14 (34.2)	4 (10.3)	0.015
No symptoms	14 (32.6)	5 (14.3)	0.062	12 (29.3)	4 (10.3)	0.049
Hypertension	38 (88.4)	23 (65.7)	0.015	31 (75.6)	34 (87.2)	0.185
Ischemic heart disease	31 (72.1)	10 (28.6)	<0.001	28 (68.3)	12 (30.8)	<0.001
Diabetes	29 (67.4)	14 (40.0)	0.015	30 (73.2)	18 (46.2)	0.014
Heart failure	28 (33.3)	5 (6.8)	<0.001	9 (22.0)	2 (5.1)	0.048
Stroke/TIA	20 (46.5)	0 (0.0)	<0.001	18 (43.9)	4 (10.3)	0.001
Six or more hospital stays	36 (73.7)	5 (14.3)	<0.001	26 (63.4)	3 (7.7)	<0.001
VKA anticoagulant treatment	23 (53.5)	5 (14.3)	<0.001	24 (58.5)	15 (38.5)	0.073
Controls the INR indicator	42 (97.7)	35 (100)	1.000	35 (85.4)	33 (84.6)	0.826
Checks NIR at least every two months	21 (48.8)	11 (31.4)	0.186	13 (31.7)	14 (35.9)	0.873
Bleeding as a result of treatment	18 (41.9)	7 (20.0)	0.040	23 (56.1)	13 (33.3)	0.041
Anxiety (HADS-A > 10)	35 (81.4)	4 (11.4)	<0.001	36 (90.0)	10 (27.8)	<0.001
Depression (HADS-D > 10)	32 (82.1)	0 (0.0)	<0.001	29 (82.9)	2 (7.7)	<0.001
Sleep disturbance (AIS > 10)	41 (95.4)	16 (45.7)	<0.001	39 (97.5)	12 (30.8)	<0.001

**Table 6 jcm-10-00011-t006:** Frailty syndrome (FS) predictors in patients with atrial fibrillation (AF) and odds ratio (OR) and logistic regression coefficients (b and beta) for FS and the sociodemographic and clinical characteristics analyzed.

Parameters (Predictors)	Frailty Syndrome	*p*-Value	OR (95% CI)	Single-Factor Analysis	Multiple-Factor Analysis
Yes (*n* = 84)	No (*n* = 74)	*b*	*p*	*beta*	*p*
*n* (%)	*n* (%)
Age > 66 years	78 (92.9)	19 (25.7)	<0.001	37.6 (14.1–100)	1.814	<0.001	-	ns
No higher education	83 (98.8)	43 (58.1)	<0.001	59.8 (7.90–453)	2.046	<0.001	-	ns
Not working (disability, retirement pension)	78 (92.9)	26 (35.1)	<0.001	24.0 (9.21–62.5)	1.589	<0.001	-	ns
Marital status - single	53 (63.1)	23 (31.1)	<0.001	3.79 (1.96–7.35)	0.666	<0.001	-	ns
Physical work	57 (67.8)	23 (31.1)	<0.001	4.68 (2.39–9.17)	0.772	<0.001	-	ns
Lack of physical activity	68 (81.0)	6 (8.1)	<0.001	48.2 (17.8–130)	1.937	<0.001	-	ns
Duration of AF > 5 years	73 (86.9)	14 (18.9)	<0.001	28.4 (12.0–67.2)	1.674	<0.001	4.649	0.001
Dyspnea	67 (79.8)	39 (52.7)	<0.001	3.54 (1.75–7.13)	0.632	<0.001	-	ns
Fatigue	67 (79.8)	28 (37.8)	<0.001	6.47 (3.18–13.2)	0.934	<0.001	-	ns
Palpitations	44 (52.4)	64 (86.5)	<0.001	0.17 (0.08–0.38)	-0.880	<0.001	-	ns
Dizziness	53 (63.1)	19 (25.7)	<0.001	4.95 (2.50–9.81)	0.800	<0.001	-	ns
Pain in the chest	36 (42.9)	15 (20.3)	0.004	2.95 (1.45–6.02)	0.541	0.003	-	ns
Sweats	34 (40.5)	13 (17.6)	0.003	3.19 (1.52–6.69)	0.580	0.002	-	ns
Fainting	35 (41.7)	5 (6.8)	<0.001	9.86 (3.60–27.0)	1.144	<0.001	-	ns
No symptoms	26 (30.9)	9 (12.2)	0.005	3.81 (1.66–8.73)	0.587	0.006	-	ns
Ischemic heart disease	59 (70.2)	22 (29.7)	<0.001	5.58 (2.82–11.1)	0.859	<0.001	-	ns
Diabetes	59 (70.2)	32 (43.2)	0.001	3.10 (1.61–5.97)	0.565	<0.001	-	ns
Heart failure	28 (33.3)	5 (6.8)	<0.001	6.90 (2.50–19.0)	0.966	<0.001	-	ns
Stroke/TIA	38 (45.2)	4 (5.4)	<0.001	14.5 (4.83–43.2)	1.336	<0.001	-	ns
Six or more hospital stays	62 (73.8)	8 (10.8)	<0.001	23.2 (9.61–56.1)	1.573	<0.001	-	ns
VKA anticoagulant treatment	47 (56.0)	20 (27.0)	<0.001	3.43 (1.76–6.70)	0.616	<0.001	-	ns
Bleeding as a result of treatment	41 (48.8)	20 (27.0)	0.005	2.57 (1.32–5.02)	0.473	0.006	-	ns
Anxiety (HADS-A > 10)	71 (85.5)	14 (19.7)	<0.001	23.4 (10.2–53.7)	1.591	<0.001	2.4727	0.036
Depression (HADS-D > 10)	61 (82.4)	2 (3.4)	<0.001	95.5 (21.6–421)	2.448	<0.001	5.5712	<0.001
Sleep disturbance (AIS > 10)	80 (96.4)	28 (37.8)	<0.001	32.9 (10.8–99.6)	2.448	<0.001	4.1052	0.014
Depression/anxiety and sleep disturbance	45 (53.6)	1 (1.3)	<0.001	495 (532–4605)				
Sleep disturbance alone	35 (41.7)	29 (39.2)	<0.001	13.3 (4.26–41.3)				
Non-depression/anxiety and non-sleep disturbance	4 (4.7)	44 (59.5)	<0.001	1.00 (ref)

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
