# Peer review of "The Role of Sleep Disturbance, Depression and Anxiety in Frail Patients with AF–Gender Differences"

_jcm, 2020, doi:10.3390/jcm10010011_

Round 1
Reviewer 1 Report
Thank you for giving me the opportunity to review this very interesting manuscript. Beata Jankowska-Polańska et al have interviewed 158 hosiptalised patients with atrial fibrillation (age >60) and assessed prevalence of sleep disturbance, depression, anxiety, and frailty in this population. The prevalence compared between men and women. The level of depression and anxiety assessed as well. 48.1% had anxiety disorders, whereas 39.9% had depression. Patients with frailty were significant associated with higher risk of sleep disturbance, depression, and anxiety.
I found the study very interesting, but it deserves a better presentation. I understood the main finding in the abstract but the main text was very unclear. The introduction needs to be shorter and to describe what is the lack of knowledge between sleep disturbance, depression, and anxiety in frail patients with AF. The methods need a more organized presentation. I was unable to understand all the analyses done in this study. There is a great confusion in the text. Results section needs to be added. The discussion section is diffusing instead of explaining the results.
I tried to sort out some concerns that need to addressed as following:
- Line 22. The authors state that depression, sleep disturbances, anxiety, and AF duration of more than 5 years predict frailty. The results, in the abstract, do not justify this statement. I wonder if the case is the opposite, namely that frailty is a risk factor for depression and anxiety.
- There is no connexion between the introduction and the aim. It is therefore difficult to understand why the authors chose to conduct this study. The lines 47 to 78 need to be reassessed.
- The authors do not describe where and how they found the studied population of 158 patients with AF. How did the authors conduct the inclusion? For instance, all patients hospitalized in a certain clinic because of atrial fibrillation during a specified period of time asked to participate? How many patients were asked to participate?
- Line 83. I believe that the aim was to assess the prevalence and not the incidence. Did I understand wrong?
- Line 97. Anxiety requiring pharmacological treatment was not an exclusion criterium?
- Line 186. “In the group of patients with AF,…” Did not this study had only one group where everybody had AF?
- Line 213. “Moreover, in the two groups studied,…” Which were those groups. The whole manuscript lacks a clear presentation of the analyses conducted.
- Section 2.5 is very confusion. A lot of these results should be presented only in tables. Furthermore, the presentation of the results needs better organization. It is difficult for the reader to understand the results. For instance, consider using subtitles for anxiety, depression, sleep and frailty. Maybe another subtitle for the more advanced analyses at the end.
- Line 219. Odds ratio analysis does not exist. Please make clear what this analysis is.
- Line 219. Frailty syndrome is named here. What is frailty syndrome, is there a definition?
- I could not find the tables, where are they? It is not possible to review the manuscript without the tables.
- Lines 247-271. In the Discussion section the authors should discuss the findings presented in the pervious section. It is common to start this section with the main result and analyze it. The authors have chosen to present more findings and to discuss things that were not mentioned before. Consider rewriting this part of the text. The whole discussion section needs to be revised.
- Did this study assess how the patients feel at the time of the hospitalization or is a greater assessment made about how it is to live with diagnose atrial fibrillation? That is something the authors need to clarify.
- I miss a discussion about what is frailty/frailty syndrome, how do we define this and what are the difficulties with this definition.
Minor concerns:
Line 8. No need for comma after “patients”.
Line 8. No need for the digit in mean age; consider writing 70 ±8.
Line 8. Consider spelling out HADS-M.
Line 9. “…to assess the adequate parameters” does not really make sense, please consider rephrasing. Maybe to state something like “...to assess and compare anxiety, depression, and sleep disturbance between frail and non-frail patients with atrial fibrillation.”
Line 12. Please consider ending the sentence with point after “to non-frail patients” and start a new sentence with “No difference…”. That would make the text easier to read.
Line 23. Please spell out AF if you choose to use this sentence.
Line 27. Please spell out AF here as it is the first time it is used in the main text. Later in the text the authors use both “AF” and “atrial fibrillation”. It would be better to stay consistent to either only AF or only atrial fibrillation.
Line 28. “Atrial fibrillation significantly affects the function of the cardiovascular system and patients’ overall functioning.” I found this sentence strange, please consider using a different expression. As for example: Atrial fibrillation is an independent risk factor of stroke, heart failure, mortality, and lower quality of life.
Line 31. Is it more often or less often?
Line 31. Maybe the authors should state “oral anticoagulation in order to prevent thromboembolic outcomes”.
Lines 31-35. “The symptoms of AF, consequences of treatment and the paroxysmal form of the disease appear to be associated with the fear of iatrogenic harm, as well as the lack of confidence in the evidence of benefit in the older population [3,4]. Patients with atrial fibrillation suffer both directly and indirectly as a result of changes in hemodynamics [5]” These sentences do not add something to the introduction, I suggest to the authors to skip those lines. That would make the text easier to read.
Line 44. Please spell out HRQoL.
Line 69. I miss a reference after “…the pathophysiology of frailty in older adults.”.
Line 73. “The population of elderly AF patients is particularly at risk of sleep disturbance, anxiety, depression and frailty syndrome.” Why?
Line 79. “In view of the above, it is necessary to assess the level of sleep disturbance and depressive symptoms in atrial fibrillation patients as well as the link between the disturbances and the prevalence of frailty syndrome.” I believe that the whole paragraph before this line does not justify that statement. The text between the lines 57 to 78 is about sleep disturbance (and frailty) in the elderly in general and not at all about a possible connexion to atrial fibrillation. I would prefer to read a short introduction about what we know about frailty-sleep disturbance in general, in the elderly and then in atrial fibrillation.
Lines 88-90. These lines belong to the results and not to the Material and methods section. By the way there is no results section. Furthermore, the material and methods should be a separated section and not one with the aim section.
Line 107. Please spell out QoL while first used here.
Line 129. Please use a headline/section for the statistical analysis.
Line 154. The whole 2.5 section belongs to the results.
Line 165. Consider rephrasing this sentence, for instance “...than male patients were diagnosed with heart failure and hyperthyroidism.” The values do not seem to be correct. (128.2 vs 13.6 p=0.041)? 128.2 participants, is that possible? Later in the same paragraph the authors use percentage, Maybe would be more appropriate with number of patients
Lines 166-169. The INR data is not of importance and the authors can skip these lines.
Line 405. Reference 1. I believe that this reference does not justify the statement in the text.
Line 448. Reference 19. It is better to refer to the latest ESC AF guidelines from 2020 instead of those from 2016.
Author Response
Thank you very much for all the comments. The whole team worked to improve the entire manuscript. All comments allowed us to notice some wrong presentation of results which could cause some confusion. This work is an important part of the work of the entire team. We are aware of some errors, but we try to clarify and correct all comments. Of course, we are ready for any further corrections. Please also consider the large contribution of work in gathering such a large study group. We hope that all these mistakes will not be grounds for rejecting work at this stage. We have tried to make this manuscript look better, according to all comments.
The response to the reviewer 1
Thank you for giving me the opportunity to review this very interesting manuscript. Beata Jankowska-Polańska et al have interviewed 158 hospitalized patients with atrial fibrillation (age>60) and assessed prevalence of sleep disturbance, depression, anxiety, and frailty in this population. The prevalence compared between men and women. The level of depression and anxiety assessed as well. 48.1% had anxiety disorders, whereas 39.9% had depression. Patients with frailty were significant associated with higher risk of sleep disturbance, depression, and anxiety.
I found the study very interesting, but it deserves a better presentation. I understood the main finding in the abstract but the main text was very unclear.
- The introduction needs to be shorter and to describe what is the lack of knowledge between sleep disturbance, depression, and anxiety in frail patients with AF. The methods need a more organized presentation. I was unable to understand all the analyses done in this study. There is a great confusion in the text. Results section needs to be added. The discussion section is diffusing instead of explaining the results. I tried to sort out some concerns that need to addressed as following:
The introduction section was shortened and partially rewritten.
- Line 22. The authors state that depression, sleep disturbances, anxiety, and AF duration of more than 5 years predict frailty. The results, in the abstract, do not justify this statement. I wonder if the case is the opposite, namely that frailty is a risk factor for depression and anxiety.
The abstract was reorganized according the reviewers suggestions.
- There is no connexion between the introduction and the aim. It is there for difficult to understand why the authors chose to conduct this study. The lines 47 to 78 need to be reassessed.
This part was corrected according to reviewers suggestion.
- The authors do not describe where and how they found the studied population of 158 patients with AF. How did the authors conduct the inclusion? For instance, all patients hospitalized in a certain clinic because of atrial fibrillation during a specified period of time asked to participate? How many patients were asked to participate?
This part was improved according to the reviewers suggestions.
In the study period (......–....), 280 patients with AF were hospitalized in the clinic. In this group, 60 patients did not meet the inclusion criteria, and 27 refused to participate. Therefore, 193 patients were included in the study and received surveys; however, during the study, 35 patients dropped out without providing a reason or did not complete the survey correctly. The final group included 158 patients.
- Line 83. I believe that the aim was to assess the prevalence and not the incidence. Did I understand wrong?
The term prevalence was used.
- Line 97. Anxiety requiring pharmacological treatment was not an exclusion criterium?
It was our mistake, the authors do thank for this correction. Of course it was the case. The final version of the manuscript was corrected.
- Line 186. “In the group of patients with AF,…” Did not this study had only one group where everybody had AF?
The sentence was improved.
- Line 213. “Moreover, in the two groups studied,…” Which were those groups. The whole manuscript lacks a clear presentation of the analyses conducted.
The sentence was improved.
- Section 2.5 is very confusion. A lot of these results should be presented only in tables. Furthermore, the presentation of the results needs better organization. It is difficult for the reader to understand the results. For instance, consider using subtitles for anxiety, depression, sleep and frailty. May be another subtitle for the more advanced analyses at the end.
The subheadings were incorporated in the results section
- Line 219. Odds ratio analysis does not exist. Please make clear what this analysis is.
Performed.
- Line 219. Frailty syndrome is named here. What is frailty syndrome, is there a definition?
The definition of frailty syndrome was improved in the introduction.
- I could not find the tables, where are they? It is not possible to review the manuscript without the tables.
The tables were embodied in the main text.
- Lines 247-271. In the Discussion section the authors should discuss the findings presented in the pervious section. It is common to start this section with the main result and analyze it. The authors have chosen to present more findings and to discuss things that were not mentioned before. Consider rewriting this part of the text. The whole discussion section needs to be revised.
The discussion was improved.
- Did this study assess how the patients feel at the time of the hospitalization or is a greater assessment made about how it is to live with diagnose atrial fibrillation? That is something the authors need to clarify.
The appropriate changes were performed.
- I miss a discussion about what is frailty/frailty syndrome, how do we define this and what are the difficulties with this definition.
The definition of frailty syndrome was incorporated and discussed.
Minor concerns:
- Line 8. No need for comma after “patients”.
Corrected
- Line 8. No need for the digit in mean age; consider writing 70 ±8.
Corrected
- Line 8. Consider spelling out HADS-M.
Corrected
- Line 9. “…to assess the adequate parameters” does not really make sense, please consider rephrasing. Maybe to state something like “...to assess and compare anxiety, depression, and sleep disturbance between frail and non-frail patients with atrial fibrillation.”
Changed
- Line 12. Please consider ending the sentence with point after “to non-frail patients” and start a new sentence with “No difference…”. That would make the text easier to read.
Changed
- Line 23. Please spell out AF if you choose to use this sentence.
Changed
- Line 27. Please spell out AF here as it is the first time it is used in the main text. Later in the text the authors use both “AF” and “atrial fibrillation”. It would be better to stay consistent to either only AF or only atrial fibrillation.
The abbreviation was implemented correctly within the whole text.
- Line 28. “Atrial fibrillation significantly affects the function of the cardiovascular system and patients’ overall functioning.” I found this sentence strange, please consider using a different expression. As for example: Atrial fibrillation is an independent risk factor of stroke, heart failure, mortality, and lower quality of life.
Corrected.
- Line 31. Is it more often or less often?
Corrected.
- Line 31. Maybe the authors should state “oral anticoagulation in order to prevent thromboembolic outcomes”.
Corrected.
- Lines 31-35. “The symptoms of AF, consequences of treatment and the paroxysmal form of the disease appear to be associated with the fear of iatrogenic harm, as well as the lack of confidence in the evidence of benefit in the older population [3,4]. Patients with atrial fibrillation suffer both directly and indirectly as a result of changes in hemodynamics [5]” These sentences do not add something to the introduction, I suggest to the authors to skip those lines. That would make the text easier to read.
Deleted
- Line 44. Please spell out HRQoL.
The appropriate change was made
- Line 69. I miss a reference after “…the pathophysiology of frailty in older adults.”.
Corrected
- Line 73. “The population of elderly AF patients is particularly at risk of sleep disturbance, anxiety, depression and frailty syndrome.” Why?
Corrected we tried to explain the mechanism of AF and sleep disorders and depression
- Line 79. “In view of the above, it is necessary to assess the level of sleep disturbance and depressive symptoms in atrial fibrillation patients as well as the link between the disturbances and the prevalence of frailty syndrome.” I believe that the whole paragraph before this line does not justify that statement. The text between the lines 57 to 78 is about sleep disturbance (and frailty) in the elderly in general and not at all about a possible connexion to atrial fibrillation. I would prefer to read a short introduction about what we know about frailty-sleep disturbance in general, in the elderly and then in atrial fibrillation.
Changed, we hope to be more clarifying
- Lines 88-90. These lines belong to the results and not to the Material and methods section. By the way there is no results section. Furthermore, the material and methods should be a separated section and not one with the aim section.
The subheadings implemented and those lines were moved to the results section according to reviewers’ suggestions
- Line 107. Please spell out QoL while first used here.
The abbreviation was explained
- Line 129. Please use a headline/section for the statistical analysis.
The subheading was inserted
- Line 154. The whole 2.5 section belongs to the results.
The 2.5 section was moved to results and renamed as 3.1
- Line 165. Consider rephrasing this sentence, for instance “...than male patients were diagnosed with heart failure and hyperthyroidism.” The values do not seem to be correct. (128.2 vs 13.6 p=0.041)? 128.2 participants, is that possible? Later in the same paragraph the authors use percentage, May be would be more appropriate with number of patients
The incorrectness was improved
- Lines 166-169. The INR data is not of importance and the author scan skip these lines.
Deleted
- Line 405. Reference 1. I believe that this reference does not justify the statement in the text.
The more suitable reference was used:
Annoni G, Mazzola P. Real-world characteristics of hospitalized frail elderly patients with atrial fibrillation: can we improve the current prescription of anticoagulants? J Geriatr Cardiol 2016; 13: 226–32.
- Line 448. Reference 19. It is better to refer to the latest ESC AF guidelines from 2020 instead of those from 2016.
The reference from 2020 was used:
Hindricks G, Potpara TS, et al. 2020 ESC Guidelines for the diagnosis and management of atrial fibrillation developed in collaboration with the European Association of Cardio-Thoracic Surgery (EACTS). Eur Heart J 2020. doi/10.1093/eurheartj/ehaa612
Reviewer 2 Report
Thank you for asking me to review this manuscript.
In this manuscript, the authors studied anxiety, depression, sleep disturbance, and frailty in patients with atrial fibrillation. A particular focus was on any gender differences in these factors. A cohort of 158 hospitalised patients with AF were included. Validated scales were used to assess anxiety, depression, sleep disturbance and frailty.
Unfortunately, a number of limitations restricts the suitability of this manuscript for publication. To name just a few: the quality of English writing renders it difficult if not impossible to comprehend the point being conveyed in many sentences. The aims appear to differ between different sections of the manuscript. Methods are incomplete, including crucial details such as the source of the study population and references for measurement scales. Results are unclear and unable to be interpreted.
As a result, this manuscript would require significant revision and rewriting to make it suitable for repeat review.
Author Response
Thank you very much for all the comments. The whole team worked to improve the entire manuscript. All comments allowed us to notice some wrong presentation of results which could cause some confusion. This work is an important part of the work of the entire team. We are aware of some errors, but we try to clarify and correct all comments. Of course, we are ready for any further corrections. Please also consider the large contribution of work in gathering such a large study group. We hope that all these mistakes will not be grounds for rejecting work at this stage. We have tried to make this manuscript look better, according to all comments.
The response to the reviewer 2
Open Review
(x) I would not like to sign my review report
( ) I wouldlike to sign my review report
English language and style
(x) Extensiveediting of English language and style required
( ) Moderate English changesrequired
( ) English language and style are fine/minor spellcheckrequired
( ) I don'tfeelqualified to judgeabout the English language and style
Thank you for asking me to review this manuscript.
In this manuscript, the authors studied anxiety, depression, sleep disturbance, and frailty in patients with atrial fibrillation. A particular focus was on any gender differences in these factors. A cohort of 158 hospitalised patients with AF were included. Validated scales were used to assess anxiety, depression, sleep disturbance and frailty.
Unfortunately, a number of limitations restricts the suitability of this manuscript for publication. To name just a few: the quality of English writing renders it difficult if not impossible to comprehend the point being conveyed in many sentences.
The new version of the manuscript was carefully checked for any language flaws by a professional translator.
The aims appear to differ between different sections of the manuscript. Methods are incomplete, including crucial details such as the source of the study population and references for measurement scales. Results are unclear and unable to be interpreted.
The suggested issues were improved in the new version of the manuscript.
As a result, this manuscript would require significant revision and rewriting to make it suitable for repeat review.
with kind regards on behalf of the authors
Beata Jankowska-Polańska
Round 2
Reviewer 1 Report
Thank you for this remarkable improved manuscript. There is still an issue I would like to address. This cross-sectional study presents a prove of association, not causality. The authors use the term “risk factor”, which is often used to declare causality, and that can lead to some misunderstanding. I propose to look over those statements (lines 31, 231, 243, 245, and 319) as I mention bellow. Otherwise I only pointed out some minor (and easy to correct) issues. I think that the manuscript can be good for acceptance if the authors provide the following corrections.
Line 31. I believe that the study can not prove causality, but rather association between depression, sleep disturbances and anxiety, and frailty syndrome. The statement “Independent predictors of depression,…” can not be justified. Please consider using another expression, as for instance: “Frailty in patients with AF is associated with higher risk of depression, sleep disturbances and anxiety.” That would be a more correct statement.
Line 231. The univariate analysis showed that high levels of anxiety, depression and sleep disturbance are associated with higher risk of frailty syndrome in the AF population. Not to confuse with risk factor of frailty syndrome, the study is not designed to show causality but association. Please rephrase.
Line 243. Those are factors associated with Frailty Syndrome and not “Risk factors for frailty syndrome in patients with AF and Chi-square test results”.
Line 245. I believe that there is not such a think as “odds ratio analysis”. Please state what kind of analysis is conducted in order to estimate the odds ratio. Regressions analysis?
Line 319. I do not think that “the results indicate furthermore that all the above mentioned factors contribute to the development of frailty syndrome or increase its severity”. The study does not prove causality but association. I would prefer to see a more cautious statement.
Minor suggestions:
Line 50. Please explain FS here as Frailty syndrome (FS). Thereafter the authors use FS and frailty syndrome alternately. I would prefer some consistency to either FS or frailty syndrome.
Line 184. Figure legend is missing.
Lines 200-208. The whole text is repeated (in a better form) in lines 211-218. Please erase the text in line 200-208.
Lines 220. It is common praxis the figures and tables to be able to stand alone without the main text. Therefore, it is needed to spell out AF and FS beneath the table. The same applies for Table 6.
Line 246. Please spell out Odds Ratio first time used in the text.
Line 252. A more cautious heading would be “possible predictors of frailty syndrome”.
Line 389-396. These lines are repeated above. It is better to erase those lines.
Author Response
Thank you for this remarkable improved manuscript. There is still an issue I would like to address. This cross-sectional study presents a prove of association, not causality. The authors use the term “risk factor”, which is often used to declare causality, and that can lead to some misunderstanding. I propose to look over those statements (lines 31, 231, 243, 245, and 319) as I mention bellow. Otherwise I only pointed out some minor (and easy to correct) issues. I think that the manuscript can be good for acceptance if the authors provide the following corrections.
The authors would like to express deep satisfaction on the encouraging words of the reviewer. Of course, we agree, that the study was not design to and does not have the power to prove causal relationship and we fully understand its limitations. Nevertheless, we are glad the study found the interest of the reviewer.
Line 31. I believe that the study can not prove causality, but rather association between depression, sleep disturbances and anxiety, and frailty syndrome. The statement “Independent predictors of depression,” can not be justified. Please consider using another expression, as for instance: “Frailty in patients with AF is associated with higher risk of depression, sleep disturbances and anxiety.” That would be a more correct statement.
The statement in question was corrected accordingly.
Line 231. The univariate analysis showed that high levels of anxiety, depression and sleep disturbance are associated with higher risk of frailty syndrome in the AF population. Not to confuse with risk factor of frailty syndrome, the study is not designed to show causality but association. Please rephrase.
The statement was corrected according to the suggestions of the reviewer.
Line 243. Those are factors associated with Frailty Syndrome and not “Risk factors for frailty syndrome in patients with AF and Chi-square test results”.
We agree, the heading was corrected.
Line 245. I believe that there is not such a think as “odds ratio analysis”. Please state what kind of analysis is conducted in order to estimate the odds ratio. Regressions analysis?
Our statistician acknowledged his mistake. The analysis was corrected accordingly.
Line 319. I do not think that “the results indicate furthermore that all the above mentioned factors contribute to the development of frailty syndrome or increase its severity”. The study does not prove causality but association. I would prefer to see a more cautious statement.
The sentence was ‘softened’ accordingly.
Minor suggestions:
Line 50. Please explain FS here as Frailty syndrome (FS). Thereafter the authors use FS and frailty syndrome alternately. I would prefer some consistency to either FS or frailty syndrome.
The change was made.
Line 184. Figure legend is missing.
The figure legend was added.
Lines 200-208. The whole text is repeated (in a better form) in lines 211-218. Please erase the text in line 200-208.
The redundant text was erased.
Lines 220. It is common praxis the figures and tables to be able to stand alone without the main text. Therefore, it is needed to spell out AF and FS beneath the table. The same applies for Table 6.
The suggested correction was performed.
Line 246. Please spell out Odds Ratio first time used in the text.
The change was made.
Line 252. A more cautious heading would be “possible predictors of frailty syndrome”.
The heading was corrected accordingly
Line 389-396. These lines are repeated above. It is better to erase those lines.
The repeated text was deleted.
